# Effective smMIPs-Based Sequencing of Maculopathy-Associated Genes in Stargardt Disease Cases and Allied Maculopathies from the UK

**DOI:** 10.3390/genes14010191

**Published:** 2023-01-11

**Authors:** Benjamin Mc Clinton, Zelia Corradi, Martin McKibbin, Daan M. Panneman, Susanne Roosing, Erica G. M. Boonen, Manir Ali, Christopher M. Watson, David H. Steel, Frans P. M. Cremers, Chris F. Inglehearn, Rebekkah J. Hitti-Malin, Carmel Toomes

**Affiliations:** 1Leeds Institute of Medical Research, University of Leeds, St James’s University Hospital, Leeds LS9 7TF, UK; 2Department of Human Genetics, Radboud University Medical Center, 6525 GA Nijmegen, The Netherlands; 3Donders Institute for Brain, Cognition and Behaviour, Radboud University Medical Center, 6525 GA Nijmegen, The Netherlands; 4Department of Ophthalmology, St. James’s University Hospital, Leeds LS9 7TF, UK; 5North East and Yorkshire Genomic Laboratory Hub, Central Lab, St. James’s University Hospital, Leeds LS9 7TF, UK; 6Sunderland Eye Infirmary, Sunderland SR2 9HP, UK; 7The Bioscience Institute, Newcastle University, Newcastle upon Tyne NE2 4HH, UK

**Keywords:** maculopathies, *ABCA4*, Stargardt, smMIPs, inherited retinal diseases, NGS

## Abstract

Macular dystrophies are a group of individually rare but collectively common inherited retinal dystrophies characterised by central vision loss and loss of visual acuity. Single molecule Molecular Inversion Probes (smMIPs) have proved effective in identifying genetic variants causing macular dystrophy. Here, a previously established smMIPs panel tailored for genes associated with macular diseases has been used to examine 57 UK macular dystrophy cases, achieving a high solve rate of 63.2% (36/57). Among 27 bi-allelic STGD1 cases, only three novel *ABCA4* variants were identified, illustrating that the majority of *ABCA4* variants in Caucasian STGD1 cases are currently known. We examined cases with *ABCA4*-associated disease in detail, comparing our results with a previously reported variant grading system, and found this model to be accurate and clinically useful. In this study, we showed that *ABCA4*-associated disease could be distinguished from other forms of macular dystrophy based on clinical evaluation in the majority of cases (34/36)

## 1. Introduction

Inherited retinal diseases (IRDs) are a heterogeneous group of individually rare but collectively common Mendelian disorders, which impact retinal function and result in vision impairment. They can be broadly classified into those conditions that impact cones first, resulting in an initial loss of central vision and visual acuity, such as cone-rod dystrophy, and those that impact rods first, resulting in an initial loss of peripheral vision and night blindness, such as in retinitis pigmentosa.

Recently, the advent of widespread next generation sequencing (NGS) has revolutionised the genetic diagnosis of patients with IRDs. For most routine research and diagnostic sequencing, enrichment for target sequences is typically performed to optimise the cost-benefits of reduced sequencing, analysis and data storage against the risk of missing causal variants in a few cases. Typically, DNA isolated from individuals with a suspected IRD will be sent for sequencing via a custom gene panel approach [1,2,3,4,5,6,7,8,9,10,11,12]. In cases where this is unsuccessful, a broader approach, such as whole exome (WES) or whole genome sequencing (WGS), will then be employed [13,14,15].

While the interpretation of detected variants remains a significant challenge, this approach has been hugely successful. Targeted sequencing of the coding regions of IRD-associated genes can provide a genetic explanation for up to two-thirds of cases with IRDs [16,17,18]. As such, significant investment has been made in developing new approaches to enrich the initial target sequences. A technology that has been very successful is based on the use of Molecular Inversion Probes (MIPs) [19,20,21], and more recently, single molecule Molecular Inversion Probes (smMIPs) [22].

MIPs allow polymerase chain reaction (PCR) based enrichment and consist of two probe arms connected by a common linker sequence. Because of the combined specificity of two times 20 nt annealing sequences, thousands of MIPs can be used simultaneously to amplify many loci without interfering with each other. Additionally, the probes can be indexed, allowing for the pooling of multiple patients in a single sequencing run. As such, this approach can be scaled to capture many loci in many cases at a comparatively low cost per sample [19,23]. MIPs-based approaches offer several advantages, such as low per-sample cost, a robust semi-automatable workflow and low sample input requirements, which make them well suited to screen large cohorts. MIPs-based approaches typically generate good coverage per sample, allowing confident variant calls [24]. Illumina recommends a coverage of 30–50× for WGS and 100× for WES (https://emea.illumina.com/science/technology/next-generation-sequencing/plan-experiments/coverage.html, accessed 14 June 2022). In contrast, MIPs-based panels can generate a mean nucleotide coverage in excess of 400× [23]. smMIPs are a further advancement of MIPs-based capture, combining scalable multiplexing with ‘single molecule tagging’. Each smMIP probe contains a unique molecular tag, which is amplified with the target sequence, allowing for tagging of source DNA molecules. In doing so, PCR duplicates can be excluded, giving higher confidence consensus calling on a per molecule basis [25]. 

MIPs, and later smMIPs, have been used successfully to study a variety of disorders, such as nephronophthisis [20], stillbirth [26], Parkinson’s disease [27] and numerous cancers, including childhood brain tumours, endometrial cancer and paediatric leukaemia [28,29,30]. Additionally, they show promise in the study of pharmacogenomics [31].

An area in which smMIPs have been used to great effect is the study of IRDs, and in particular *ABCA4*-associated disease. This is the most common single gene underlying IRD as well as an exemplar model gene for other IRDs and Mendelian conditions. smMIPs have been used to provide a molecular diagnosis to patients with *ABCA4*-associated Stargardt disease (STGD1) and to capture the entirety of *ABCA4* in large cohorts of patients at a high depth of coverage, up to an average of ~700× in large cohorts [18,32]. This robust sequencing strategy has revealed the complex genetic architecture of *ABCA4*-associated disease, including the effect of deep intronic variants, common low-penetrance variants, copy number variants (CNVs) and complex alleles [18,33,34]. The study of previously published *ABCA4* variants led to the conclusion that up to a quarter of *ABCA4*-associated disease displays complex genetic architecture, where together with two *ABCA4* alleles, additional genetic or non-genetic factors play a significant role in the development of late-onset *ABCA4*-associated STGD1[34]. Additionally, this in-depth study of *ABCA4* facilitated the classification of *ABCA4* variants into ‘Benign, Mild, Moderately Severe and Severe’ categories based on previous functional and clinical studies and comparisons of frequencies in affected and non-affected populations [34]. A phenotype–genotype model has been proposed by Cremers et al. by comparing STGD disease trajectories associated with *ABCA4* variants [35]. In this model, disease trajectory is defined by the spatial extent of the disease at a given age. Broadly, this model defines four main stages of disease progression: ‘Macular’ (effects confined to the central macula), ‘Extramacular’ (disease effects visible beyond arcades and regions nasal to the optic disk), ‘Transitional’ (disease effects confluent across the posterior pole initiating peripheral involvement and outer retinal atrophy), and ‘Advanced’ (evidence of multiple lesions and across the coalesce across the posterior pole). 

In this study, we have utilised an smMIPs panel created to capture the coding sequences and selected non-coding regions in 105 genes and loci associated with macular diseases (MDs) [23] to interrogate a cohort of genetically unexplained UK macular cases and to investigate phenotype–genotype correlations. In doing so, we aim to provide a genetic diagnosis to these patients and their families, which in turn may inform potential future genetic therapies.

## 2. Materials and Methods

### 2.1. smMIPs Design

All smMIPs were designed as part of an ‘MD-smMIPs panel’ to capture the coding regions of 105 genes and non-coding or regulatory loci associated with inherited MD and age-related MD (AMD), known sites of pseudo-exons, and the mitochondrial genome. This totalled 17,394 smMIPs each of which targeted a 225 nucleotide locus. The smMIPs pool used by Hitti-Malin and colleagues [23] was used in this study. The full list of loci targeted by Hitti-Malin and colleagues can be viewed at [23].

### 2.2. Patient Cohort

All patients were diagnosed and recruited to the study by ophthalmologists at hospitals in the North of England. Blood samples were collected from patients and family members after obtaining informed consent. Ethical approval was provided by the Leeds East Teaching Hospitals NHS Trust Research Ethics Committee (Project number 17/YH/0032). DNA was isolated using standard protocols by Yorkshire Regional Genetics (Leeds, UK). Patients were selected based on a clinical diagnosis of STGD, a STGD-like phenotype, macular degeneration or retinal dystrophy characterised by primarily macular or cone involvement based on symptoms, family history, clinical examination, retinal imaging and, when indicated, ocular electrophysiology. Images for this text were collected using Ultra Wide Field Autofluorescence (Leeds, UK). 

### 2.3. Sample Preparation

Genomic DNA samples were quantified and diluted to a DNA concentration of 15–25 nanograms per microliter (ng/μL). Sample quality was determined by gel electrophoresis of 100 ng of patient DNA sample with a 1 kilobase plus ladder (Invitrogen, Paisley, UK) and Lambda DNA-HindIII marker (Thermo Fisher Scientific, Waltham, MA, USA). Samples with sufficient DNA integrity of both low and high molecular weight proceeded to library preparation. DNA libraries were prepared for each proband using the High Input DNA Capture Kit, Chemistry 2.3.0H produced by Molecular Loop Biosciences Inc (Woburn, MA, USA), according to manufacturer guidelines. Further information on library preparation of samples and sequencing has been published elsewhere [23].

### 2.4. Variant Calling and Annotation

Details on variant calling parameters and annotation of variants have been previously described elsewhere [32]. In brief, raw FASTQ files were quality control filtered and the random identifiers were trimmed and stored with the read identifier for downstream use. The paired end reads were then mapped to the human reference genome version hg19 using BWA mem (v0.7.12). The ligation and extension arm of the smMIPs as well as overlap between read pairs were trimmed following alignment. Duplicated reads were also removed based on the random identifiers. Remaining reads were grouped based on patient barcodes and separated into a binary alignment map (BAM) file per patient. Variants were called using GATK (v3.4-46). 

### 2.5. Variant Prioritisation

Variants were visualised in patient BAM files using Integrative Genome Viewer (IGV) v2.7.2 [22]. Variant analysis was performed in four stages. In the first stage, CNV analysis was performed using an Excel script comparing per smMIP coverage to normalised coverage across all samples and in the sample of interest, as described elsewhere [23,36]. In the second stage, previously published variants in *ABCA4*, obtained from the *ABCA4* Leiden Open (source) Variation Database (LOVD) [37], were extracted, including common variants known to be pathogenic and deep intronic variants [38] (LOVD; https://databases.lovd.nl/shared/genes/ABCA4; accessed on 10 June 2022). In the third and fourth stages, homozygous and heterozygous rare variants (minor allele frequency (MAF) ≤ 0.5% and 0.1%, respectively) were prioritised based on predicted protein effect and in silico pathogenicity predictions. PhyloP (≥2.7), Grantham (≥80) and CADD-PHRED (≥15) were used [39,40,41]. ACMG scores were obtained through Franklin by Genoox (https://franklin.genoox.com, accessed on 15th June 2022). Only class 3, 4 or 5 variants according to ACMG guidelines [42] (i.e., VUS, likely pathogenic or pathogenic, respectively) were considered. Cases were considered “very likely solved”, “possibly solved” or “unsolved” according to previously utilised grading criteria in keeping with ACMG guidelines [42]. As phasing by segregation was not possible for most probands, probands with two different rare variants found in the same gene were assumed to carry these variants in *trans* (except in the case of known complex alleles). However, without phasing information it is not possible to exclude the possibility the variants are in *cis* and there is a third, undiscovered pathogenic variant in trans. As such, cases were considered very likely or possibly solved to reflect this uncertainty.

To ensure the quality of the data, a selection of variants (17/84, 20%) were confirmed by independent PCR amplification and sequencing using either Sanger sequencing or Oxford Nanopore sequencing.

## 3. Results

Using the MD-smMIPs panel [23], 105 loci associated with inherited MDs and/or AMD, 60 published variants and eight unpublished deep intronic variants [G. Arno, Z. Corradi, F.P.M. Cremers, unpublished data) were sequenced in a cohort of 57 UK MD cases. Thirty-five cases (60%) were considered very likely solved, one was possibly solved and 21 remained genetically unsolved. Solved cases with their identified variants are listed in Table 1. All variants which were tested by independent amplification and sequencing were confirmed to be present. As phasing by segregation was not possible for most cases, rare compound heterozygous variants were assumed to be in *trans*, acknowledging this is an assumption 

The majority of cases were genetically explained by variants in *ABCA4* (75%). Variants in other genes represented 25% of the solved cases: *PRPH2* (OMIM 179605), *BEST1* (OMIM 607854), *PROM1* (OMIM 604365), *CRB1* (OMIM 604210) and *C1QTNF5* (OMIM 608752) were represented. 

As can be seen in Table 1, of the solved cases with causative variants in *ABCA4*, 13 had two pathogenic *ABCA4* variants and eight had three pathogenic *ABCA4* variants (probands 1337, 1808, 2843, 5219, 5270, 5604, 5861, 5863). While it is assumed that in the cases with three variants, this comprised a complex allele with two of the three variants in *cis* and the final variant in *trans,* this cannot be determined with certainty without phasing of the variants. However, some known *ABCA4* variants have previously been demonstrated to almost always be in *cis* as a single complex allele. These include c.2588G>C and 5603A>T [43] and 5603A>T c.5461-10T>C [18]. In cases with these variants, it was assumed that the variants previously shown to be found as a single complex allele were in *cis*. Finally, six cases had four pathogenic *ABCA4* variants (probands 1746, 2469, 4126, 5608, 5852, 5853). Similarly, it is assumed that in these cases the most likely allele combination was two complex alleles in *trans*, though this cannot be determined without phasing of the locus. The common *ABCA4*:c.5603A>T, p.(Asn1868Ile) variant was present in a heterozygous state in 12 of the cases genetically explained by *ABCA4* variants and in a homozygous state in four of these cases. This variant was present in all of the cases with more than two *ABCA4* variants. Additionally, there were seven unsolved cases with one *ABCA4* variant (Appendix A), where c.5603A>T, p.(Asn1868Ile) was present in five of these cases. 

The phenotypes of the cases were revisited in light of the genetic results (Table 2). Of particular note, we correlated the previously calculated severity of *ABCA4* variants [44] with age at diagnosis, peripheral involvement and progression of disease, in light of the model proposed by Cremers and colleagues [35]. This was possible for 25 of 27 cases which were genetically explained by biallelic variants in *ABCA4*. We found that for 18 of the 25 cases, the genotype–phenotype relationship was as predicted by the scoring model. Three exemplar cases are displayed in Figure 1A–C. Of the 18 cases that fit with the model, eight had more than two pathogenic variants in *ABCA4*; however, these variants form a complex allele, which is composed of two variants that have been observed in *cis*, allowing the phase of these cases to be assumed with a high degree of certainty. As highlighted in Figure 1D, it is assumed that the presence of the known complex alleles leads to a good genotype–phenotype correlation. There were four cases where more than two pathogenic variants were identified and as such the phase could not be established, preventing accurate assessment using the model. This is shown in the case displayed in Figure 1E; here, phasing was not possible and therefore a correlation could not be studied. Additionally, there were two cases where the phenotype–genotype model did not fit with the clinically examined grade (Table 2). Therefore, in summary, of the 25 biallelic *ABCA4* cases with sufficient clinical information available, 19 correlated with the model and two did not, while the remaining four could not be interpreted due to lack of phasing information.

We also contrasted *ABCA4*-associated disease with phenotypes caused by variants in other genes. We compared 27 cases with *ABCA4*-associated disease to nine cases with disease caused by variants in genes other than *ABCA4*. We found that all 27 cases of *ABCA4*-associated disease were correctly identified as STGD (i.e., STGD1). For nine cases that were genetically solved by variants in other genes, only two were erroneously diagnosed as STGD. We also examined the phenotypes of the unsolved cases. Of the 21 unsolved cases, there were diagnoses of STGD (N = 4), MD (N = 6), cone-rod dystrophy (N = 10) and one diagnosis of North Carolina macular dystrophy. These cases will be taken forward for WES or WGS.

## 4. Discussion

Screening with MIPs or smMIPs-based technologies has previously been demonstrated to be cost-effective for screening large populations of patients with Mendelian and complex genetic disorders. In this study, we used a recently developed smMIPs panel for MDs to screen a cohort of genetically unexplained MD cases from the UK. With an overall diagnostic yield of 63.2%, this has proved a successful initial screening strategy, which compares favourably with other targeted exon-based enrichment strategies. It has been suggested that targeted sequencing of the coding regions of IRD-associated genes can solve up to two-thirds of IRD cases [16]. This cohort of cases recruited for primarily macular involvement was well suited for screening by this panel, leading to enrichment of STGD1 cases in particular.

Previous efforts to generate a comprehensive database of known *ABCA4* variants greatly increased the confidence in the interpretation of called variants (https://www.lovd.nl/, accessed on 10 June 2022). LOVDs have recently been completed for variants in 200 genes implicated in non-syndromic IRDs and Usher syndrome published up to 2019 or 2020 (F.P.M. Cremers, I. Fokkema, S. Roosing, J.T. den Dunnen, unpublished data). Similarly, the work to categorise the proposed severity of *ABCA4* variants [34] allowed more confident variant prioritisation. The common *ABCA4*:c.5603A>T, p.(Asn1868Ile) variant (MAF in total gnomAD: 6.6%) was found in a high proportion of cases with *ABCA4* variants (10/27 individuals). In STGD1 cases, the *ABCA4*:c.2588G>C, p.[Gly863Ala;Gly863del] variant has consistently been observed to be in *cis* with c.5603A>T, p.(Asn1868Ile), and this allele has been observed to be present in up to 50% of all *ABCA4* complex alleles, which was replicated in this cohort (six of 11 individuals with presumed complex alleles) [33,43]. Of further interest, there were four probands that shared the genotype; *ABCA4*: c.[2588G>A;5603A>T];[5461-10T>C;5603A>T]. This allele combination with a mild-complete penetrant and severe allele resulted in ‘classic’ STGD, progressing to an advanced disease stage by the 6th decade, as observed previously [33,35]. 

The classical STGD1 phenotype can commonly be distinguished from other forms of MD based on clinical examination [35]. Other forms of *ABCA4-*associated disease can still be discriminated from other forms of MD; however, this can be more challenging depending on the stage and severity of disease. Likewise, some MDs caused by variants in other genes, such as *PRPH2* and *PROM1,* are well known phenocopies of STGD1 [34,45,46,47,48]. Thorough genetic screening is useful for informing the clinical diagnosis. This is vital for accurately advising patients of likely disease progression and genetic risks for offspring.

Of interest, a single case, 3616, had pathogenic variants, which could be considered causal, in both *ABCA4* and *PRPH2*. This was a sporadic case so it was not possible to determine whether there was a dominant or recessive inheritance pattern; however, this case had a diagnosis of ‘classical STGD’. It was not possible to accurately grade this case per the model described by Cremers and colleagues [35], as the combination of variants will have an unknown effect on disease progression. As such, it would be of interest to follow up this case in the future and determine whether future disease progression is more severe than expected due to the presence of the *PRPH2* variant. 

As successful as MIPs-based strategies have been for large-scale targeted screening, they have some drawbacks. As with all short-read based sequencing, it is typically not possible to determine the phase of the variants unless there is DNA available from further family members or variants in *cis* are in close proximity and can be captured by a sequencing read pair. Alternatively, if they are close enough to be captured using regular genomic DNA PCR, the phase can also be established. Generally, if two pathogenic variants are found they are assumed to be in *trans*, and if more than two variants are found, the two variants in closest proximity to one another are assumed to be in *cis*, assuming no known complex alleles are present. However, this is only ever a ‘best guess’ and it is possible that a given case is mischaracterised due to seemingly causative biallelic variants being in *cis* rather than in *trans* as assumed. The advent of long read sequencing promises to fill this gap, but it is not yet accurate or cost effective enough for large-scale screens, such as this [49,50]. 

A possible explanation for the remaining unsolved cases following an MIPs- or other amplification-based enrichment methods is that highly repetitive or GC-rich regions are often not amenable to amplification-based enrichment, meaning these regions are not well captured and/or not effectively sequenced using NGS. For example, exon 15 of the *RPGR-ORF15* transcript is famously difficult to sequence due to repetitive and purine-rich sequence, despite accounting for up to 60% of cases with *RPGR*-associated disease [51]. *RPGR* was included as part of this screening as *ORF15* variants can result in cone-rod dystrophy, but it is not captured as efficiently as other targets. Additionally, this screen only targeted the 5′ UTR and protein coding exons and known intronic variants of the target genes. In light of this, a major benefit of smMIPs-based sequencing is that the assay can be adapted to capture new regions of interest and to incorporate updated smMIP targets to focus on pathogenic loci as they are discovered and to improve coverage of regions that prove challenging to capture. Although none were found, this smMIPs panel contains probes for 39 deep-intronic *ABCA4* variants. Conversely, an exome or clinical exome cannot quickly be tailored to target new pathogenic loci. Finally, it can be difficult to detect novel structural variation using a targeted approach that is limited to coding regions, since SV breakpoints are unlikely to be captured. 

The phenotype–genotype relationship for *ABCA4*-associated disease was examined in light of the model discussed by Cremers and colleagues [35]. From this, we conclude that the model is accurate and clinically relevant. While there were 15 cases in this analysis with more than two variants, the majority had a known complex allele (for example six cases had *ABCA4*: c.[2588G>A;5603A>T]). This allowed for grading of these cases with a high degree of confidence, even without phasing due to a lack of familial DNA. However, not all of these cases fit the model, indicating undiscovered modifiers may be at play. Additionally, five cases carried three or more variants that are not part of known complex alleles. In these cases, it is challenging to implement the model without phasing data. The presence of, as yet, uncovered complex or modifier alleles can confound the interpretation. Consequently, there are six cases that did not definitively fit within the model. For these cases, further sequencing to examine the entirety of *ABCA4* for undiscovered deep-intronic variants or modifiers could be fruitful. Additionally, long read sequencing to phase cases with no available familial DNA could be used for cases with more than two variants in particular.

From this, we conclude that targeted sequencing by smMIPs is a sensitive and cost-effective screening method in a cohort of MD cases. Comparison between *ABCA4*-associated disease and other forms of retinal dystrophy revealed that, for the majority of cases, genetic explanations obtained through this study were in agreement with the diagnosis obtained through clinical examination. As such, cost-effective sequencing such as this smMIPs panel may be useful as a genotype-first approach. Finally, we find that the genotype–phenotype model detailed in Cremers et al. [35] is robust when compared to our cohort and clinically useful for indicating likely disease progression, acknowledging that undiscovered modifiers in *ABCA4* and other genes may significantly affect progression. Further work to develop similar models for other forms of retinal dystrophy would prove useful in practice. This work provides direct benefit to patients through a genetic diagnosis for many of the cohort (pending verification in an accredited diagnostic facility) and the identification of novel disease-causing variants. Additionally, the identification of genotype–phenotype correlations allows for informing patients of likely disease progression.

## Figures and Tables

**Figure 1 genes-14-00191-f001:**
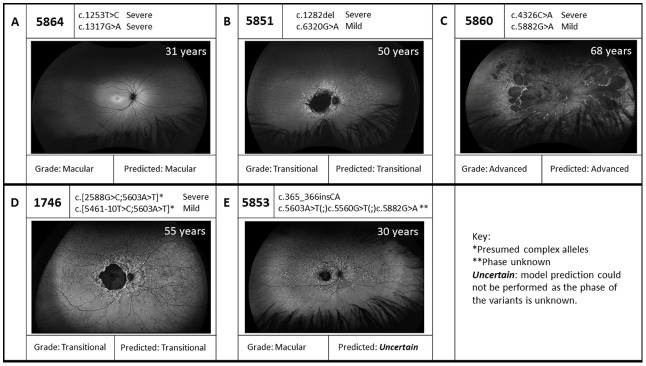
Exemplar genotype–phenotype correlations for *ABCA4*-associated disease with associated Ultra Wide Field Autofluorscence images. (**A**–**C**) show three exemplar cases which fit with the model proposed by Cremers et al. [35] based on grading by an ophthalmologist based on definitions from Cremers and colleagues. (**D**,**E**) show two cases which show the importance of accurate phasing information for predicting the result of the mutations. Case 1746 has two known complex alleles, allowing the phase to be predicted with a relatively high degree of certainty. Case 5853 has four variants which could not be phased, preventing accurate prediction.

**Table 1 genes-14-00191-t001:** Identified variants considered to very likely or possibly solve probands. Novel variants are highlighted in bold text. As no segregation analysis was performed, variants were distributed on two alleles from 5’ to 3’ considering proximity. When variants c.5461-10T>C and c.5603A>T, or variant c.2588G>C and c.5603A>T were found in the same patient, they are reported as complex alleles, as they are found in these combinations in >95% of reported STGD cases. ACMG classifications were obtained using Franklin by Genoox. P = Pathogenic, LP = Likely pathogenic, VUS = Variant of uncertain significance, LB = likely benign. # = Segregation analysis was performed.

ID	Gene	Allele 1			Allele 2		
		cDNA	Protein	ACMG	cDNA	Protein	ACMG
1337	*ABCA4*	c.5603A>T(;)5819T>C	p.(Asn1868Ile)(;)(Leu1940Pro)	VUS,LP	c.6817-2A>C	p.(?)	P
1746	*ABCA4*	c.[2588G>C;5603A>T]	p.[[Gly863Ala,Gly863del];(Asn1868Ile)]	P	c.[5461-10T>C;5603A>T]	p.[[Thr1821Aspfs*6,Thr1821Valfs*13]; (Asn1868Ile)]	P
1808	*ABCA4*	c.[5461-10T>C;5603A>T]	p.[[Thr1821Aspfs*6,Thr1821Valfs*13]; (Asn1868Ile)]	P	c.5882G>A	p.(Gly1961Glu)	P
2469	*ABCA4*	c.[2588G>C;5603A>T]	p.[[Gly863Ala,Gly863del];(Asn1868Ile)]	P	c.[5461-10T>C;5603A>T]	p.[[Thr1821Aspfs*6,Thr1821Valfs*13]; (Asn1868Ile)]	P
2843	*ABCA4*	**c.4016G>A(;)5313-1_5313del**	**p.(Cys1339Tyr)(;)(?)**	VUS,P	c.6088C>T	p.(Arg2030*)	P
3536	*ABCA4*	c.3113C>T	p.(Ala1038Val)	P	c.1906C>T	p.(Gln636*)	P
3616	*ABCA4*	c.4469G>A	p.(Cys1490Tyr)	P	c.5603A>T	p.(Asn1868Ile)	VUS
	*PRPH2*	c.623G>A	p.(Gly208Asp)	P	--	--	--
3656 #	*ABCA4*	c.4139C>T	p.(Pro1380Leu)	P	c.5882G>A	p.(Gly1961Glu)	P
4126	*ABCA4*	c.4774-27T>C(;) 5196+1137G>A	p.[=;Gly1592Alafs*113](;) [=;Met1733Glufs*78]	LB, P	c.[5461-10T>C;5603A>T]	p.[[Thr1821Aspfs*6,Thr1821Valfs*13]; (Asn1868Ile)]	P
5219	*ABCA4*	c.634C>T	p.(Arg212Cys)	P	c.[5461-10T>C;5603A>T]	p.[[Thr1821Aspfs*6,Thr1821Valfs*13]; (Asn1868Ile)]	P
5270	*ABCA4*	c.1906C>T	p.(Gln636*)	P	c.[2588G>C;5603A>T]	p.[[Gly863Ala,Gly863del];(Asn1868Ile)]	P
5349	*ABCA4*	c.1906C>T	p.(Gln636*)	P	c.5603A>T	p.(Asn1868Ile)	VUS
5604	*ABCA4*	c.4577C>T(;)4469G>A	p.(Thr1526Met)(;)(Cys1490Tyr)	P,P	c.5603A>T	p.(Asn1868Ile)	VUS
5607	*ABCA4*	c.3259G>A	p.(Glu1087Lys)	P	c.6089G>A	p.(Arg2030Gln)	P
5608	*ABCA4*	c.[2588G>C;5603A>T]	p.[[Gly863Ala,Gly863del];(Asn1868Ile)]	P	c.[5461-10T>C;5603A>T]	p.[[Thr1821Aspfs*6,Thr1821Valfs*13]; (Asn1868Ile)]	P
5609	*ABCA4*	c.4139C>T	p.(Pro1380Leu)	P	c.4139C>T	p.(Pro1380Leu)	P
5851	*ABCA4*	**c.1282del**	**p.(Val428Serfs*7)**	P	c.6320G>A	p.(Arg2107His)	P
5852	*ABCA4*	c.[2588G>C;5603A>T]	p.[[Gly863Ala,Gly863del];(Asn1868Ile)]	P	c.[5461-10T>C;5603A>T]	p.[[Thr1821Aspfs*6,Thr1821Valfs*13]; (Asn1868Ile)]	P
5853	*ABCA4*	**c.365_366insCA**	**p.(Gly123Metfs*32)**	P	c.5560G>T(;)5603A>T(;)5882G>A	p.(Val1854Leu)(;)(Asn1868Ile)(;) (Gly1961Glu)	LP,VUS,P
5854	*ABCA4*	c.4195G>A	p.(Glu1399Lys)	LP	c.5318C>T	p.(Ala1773Val)	P
5857 #	*ABCA4*	c.6229C>T	p.(Arg2077Trp)	P	c.6229C>T	p.(Arg2077Trp)	P
5860	*ABCA4*	c.4326C>A	p.(Asn1442Lys)	LP	c.5882G>A	p.(Gly1961Glu)	P
5861	*ABCA4*	c.5714+5G>A	p.[=,Glu1863Leufs*33]	P	c.[5461-10T>C;5603A>T]	p.[[Thr1821Aspfs*6,Thr1821Valfs*13]; (Asn1868Ile)]	VUS, P
5862	*ABCA4*	c.1906C>T	p.(Gln636*)	P	c.4577C>T	p.(Thr1526Met)	P
5863	*ABCA4*	c.[2588G>C;5603A>T]	p.[[Gly863Ala,Gly863del];(Asn1868Ile)]	P	c.4537del	p.(Gln1513Argfs*13)	P
5864	*ABCA4*	c.1253T>C	p.(Phe418Ser)	P	c.1317G>A	p.(Trp439*)	P
5865	*ABCA4*	c.1906C>T	p.(Gln636*)	P	c.5603A>T	p.(Asn1868Ile)	VUS
3670	*BEST1*	c.728C>T	p.(Ala243Val)	P	--	--	--
3654	*BEST1*	c.889C>T	p.(Pro297Ser)	P	--	--	--
4030	*C1QTNF5*	c.489C>G	p.(Ser163Arg)	P	--	--	--
5258	*CRB1*	c.249T>A	p.(Tyr83*)	LP	c.2506C>A	p.(Pro836Thr)	LP
3615	*PROM1*	c.1117C>T	p.(Arg373Cys)	P	--	--	--
5610	*PROM1*	c.1117C>T	p.(Arg373Cys)	P	--	--	--
3798	*PRPH2*	c.638G>A	p.(Cys213Tyr)	P	--	--	--
4767 #	*PRPH2*	c.394del	p.(Gln132Lysfs*7)	P	--	--	--
5855	*PRPH2*	c.291G>A	p.(Trp97*)	LP	--	--	--

**Table 2 genes-14-00191-t002:** Genotype–phenotype correlation for ABCA4 associated disease. Grading and model comparison from Cremers et al., 2020. Identified alleles are reported in protein notation, novel alleles are in bold text. Severity score for the variants was obtained from Cornelis et al., 2022. In “Diagnosis” columns, when not otherwise specified, STGD refers to intermediate/classical Stargardt disease.

ID	Clinical data			Genetic data					Match to Model
	Age at Grading	Grade	Diagnosis	Allele 1	Severity	Allele 2	Severity	Diagnosis	
1337	59	3	STGD	p.(Asn1868Ile)(;)(Leu1940Pro)	MildLP Severe	p.(?) †	Severe	STGD	Uncertain
1746	55	3	STGD	p.[[Gly863Ala,Gly863del]; (Asn1868Ile)]	Mild	p.[[Thr1821Aspfs*6,Thr1821Valfs*13]; (Asn1868Ile)]	Severe	STGD	Yes
1808	34	1	STGD	p.[[Thr1821Aspfs*6,Thr1821Valfs*13];(Asn1868Ile)]	Severe	p.(Gly1961Glu)	Mild	STGD	Yes
2469	36	3	STGD	p.[[Gly863Ala,Gly863del]; (Asn1868Ile)]	Mild	p.[[Thr1821Aspfs*6,Thr1821Valfs*13]; (Asn1868Ile)]	Severe	STGD	Yes
2843	22	3	CRD	**p.(Cys1339Tyr)(;)(?) †**	Unknown Severe	p.(Arg2030*)	Severe	STGD	Uncertain
3536	60	1/2	Late onset STGD	p.(Ala1038Val)	Mild	p.(Gln636*)	Severe	STGD	No
4126	13	1/2	STGD	p.[=;Gly1592Alafs*113](;) [=;Met1733Glufs*78]	Benign Mild	p.[[Thr1821Aspfs*6,Thr1821Valfs*13]; (Asn1868Ile)]	Severe	STGD	Uncertain
5219	68	4	Early onset STGD	p.(Arg212Cys)	Severe	p.[[Thr1821Aspfs*6,Thr1821Valfs*13]; (Asn1868Ile)]	Severe	Early onset STGD	Yes
5349	40	2	STGD	p.(Gln636*)	Severe	p.(Asn1868Ile)	MildLP	STGD	Yes
5604	54	4	STGD	p.(Thr1526Met)(;)(Cys1490Tyr)	Moderate Severe	p.(Asn1868Ile)	MildLP	STGD	Uncertain
5607	33	2	STGD	p.(Glu1087Lys)	Severe	p.(Arg2030Gln)	Mild	STGD	Yes
5608	45	3	STGD	p.[[Gly863Ala,Gly863del]; (Asn1868Ile)]	Mild	p.[[Thr1821Aspfs*6,Thr1821Valfs*13]; (Asn1868Ile)]	Severe	STGD	Yes
5609	37	4	STGD	p.(Pro1380Leu)	Moderate	p.(Pro1380Leu)	Moderate	STGD	No
5851	50	3	STGD	**p.(Val428Serfs*7)**	Severe	p.(Arg2107His)	Mild	STGD	Yes
5852	66	3	STGD	p.[[Gly863Ala,Gly863del]; (Asn1868Ile)]	Mild	p.[[Thr1821Aspfs*6,Thr1821Valfs*13]; (Asn1868Ile)]	Severe	STGD	Yes
5853	30	1	STGD	**p.(Gly123Metfs*32)**	Severe	p.(Val1854Leu)(;)(Asn1868Ile)(;) (Gly1961Glu)	Severe MildLP Mild	STGD	Uncertain
5854	36	2	STGD	p.(Glu1399Lys)	Mild	p.(Ala1773Val)	Severe	STGD	Yes
5857	20	3	Early onset STGD	p.(Arg2077Trp)	Severe	p.(Arg2077Trp)	Severe	Early onset STGD	Yes
5860	31	1	STGD	p.(Asn1442Lys)	Severe	p.(Gly1961Glu)	Mild	STGD	Yes
5861	37	3	STGD	p.[=,Glu1863Leufs*33]	Moderate	p.[[Thr1821Aspfs*6,Thr1821Valfs*13]; (Asn1868Ile)]	Severe	STGD	Yes
5862	17	3	Early onset STGD	p.(Gln636*)	Severe	p.(Thr1526Met)	Moderate	Early onset STGD	Yes
5863	42	2	STGD	p.[[Gly863Ala,Gly863del]; (Asn1868Ile)]	Mild	p.(Gln1513Argfs*13)	Severe	STGD	Yes
5864	68	4	STGD	p.(Phe418Ser)	Severe	p.(Trp439*)	Severe	STGD	Yes
5865	49	2	STGD	p.(Gln636*)	Severe	p.(Asn1868Ile)	MildLP	STGD	Yes
3656	42	1	Occult MD	p.(Pro1380Leu)	Moderate	p.(Gly1961Glu)	Mild	STGD	Yes

† = Non-coding variants predicted to affect splicing for which in vitro assays have not been performed. Early onset, <10 years old at onset; Intermediate, 10–40 years old at onset; late-onset, >40 year old at onset. In “Grade” column: 1 = Macular, 2 = Extramacular, 3 = Transitional, 4 = Advanced. STGD, Stargardt disease; CRD, Cone-rod dystrophy; MD, Macular dystrophy. MildLP, Mild Low Penetrance.

## Data Availability

The data that support the findings of this study are available from the corresponding author upon reasonable request.

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
