# Peer review of "Effective smMIPs-Based Sequencing of Maculopathy-Associated Genes in Stargardt Disease Cases and Allied Maculopathies from the UK"

_genes, 2023, doi:10.3390/genes14010191_

Round 1
Reviewer 1 Report
A manuscript by Benjamin Mc Clinton et al. suggesting single molecule molecular inversion Probes (smMIPs) panel as an effective detection method of maculopathy-associated genes in ABCA4-related disease and allied maculopathies from the north area of UK. Indeed, the authors were able to confirm a genetic diagnosis in previously unsolved cases in a small cohort and reached solve rate as high as over 60% and suggest SmMIPs panel as a first choicemgenetic testing for macular dystrophies. Moreover, this manuscript strengthens to some extent the applicability of previously suggested genotype-phenotype correlation model (by Cremers et al, 2020) to be useful in indicating ABCA4-retinopathy progression.
It is commonly known that interpretation of detected variants remains a significant challenge in unsolved cases. Thereby, new technologies have been applied and one of them is smMIPs. This method have been previously used in other conditions, such as nephronophtisis or cancers among others. Also, it have been already studied in large cohorts of ABCA4-related retinopathy with successful detection rate shown.
This study has few main points:
1. The manuscript is well-written and data are presented in a clear and coherent way.
2. The new suggested methodology has few advantages, such as high solve rate, possibility to screen large cohorts to a low economical costs.
3. In terms of ongoing clinical trials and need for correct genetic diagnosis, the use of new genetic technologies to increase solve rate of genetic eye conditions in clinical settings is of paramount importance.
4. This manuscript could be used as a stronger proof of concept to implement smMIPs panel into clinical practice in single institution, or being a pilot study for further research in the UK and further globally.
5. Limitations of technology are covered in the Discussion section.
Weaknesses:
1. The study sample is relatively small (taken into account the frequency of ABCA4-realted disease in general population), chosen from one area of one country only.
2. The level of novelty is moderate/low, strengthening the previously studied concept. Can be useful, however, to change current clinical practice in a geographical area of one country.
Comments:
1. Abstract, rows 28-29: “ABCA4-associated disease could be distinguished from other forms of macular dystrophy based on clinical evaluation”. It is known that ABCA4-related disease may be very heterogenous, overlapping with e.g. PRPH2-, PROM1-related retinal degeneration among others. The study cohort was relatively small and I would be careful in drawing this opinion on a broader scale, e.g. write: in this study, we showed that ABCA4-associated disease could be distinguished from other forms…
2. Introduction, rows 72-76: marge this paragraph with preceding paragraph, where MIPs have been described.
3. Results, rows 188-189: “As phasing by segregation was not possible for most cases, compound heterozygous variants were assumed to be in trans”. This assumption is a speculation. Could you, please, comment on that?
Reviewer 2 Report
This paper describes a study in which a smMIP panel was used to analyse a cohort of genetically non-designated macular disease cases.
The authors hypothesise that the smMIP analysis is an effective screening method that is cost effective, sensitive and complements whole genome sequencing. With an overall diagnostic yield of more than 60% of the previously non-described patients, the results support the standpoint that the analyses are useful. The authors conclude that this approach should serve a direct benefit for the patients.
